# AdaFuse: Adaptive Temporal Fusion Network for Efficient Action Recognition

**Yue Meng**[1]*   **Rameswar Panda**[2,3]   **Chung-Ching Lin**[4]   **Prasanna Sattigeri**[3]
**Leonid Karlinsky**[3]   **Kate Saenko**[2,5]   **Aude Oliva**[1,2]   **Rogerio Feris**[2,3]
[1]Massachusetts Institute of Technology   [2]MIT-IBM Watson AI Lab
[3]IBM Research   [4]Microsoft   [5]Boston University

## Abstract

Temporal modelling is the key for efficient video action recognition. While understanding temporal information can improve recognition accuracy for dynamic actions, removing temporal redundancy and reusing past features can significantly save computation leading to efficient action recognition. In this paper, we introduce an adaptive temporal fusion network, called AdaFuse, that dynamically fuses channels from current and past feature maps for strong temporal modelling. Specifically, the necessary information from the historical convolution feature maps is fused with current pruned feature maps with the goal of improving both recognition accuracy and efficiency. In addition, we use a skipping operation to further reduce the computation cost of action recognition. Extensive experiments on Something V1&V2, Jester and Mini-Kinetics show that our approach can achieve about 40% computation savings with comparable accuracy to state-of-the-art methods. The project page can be found at https://mengyuest.github.io/AdaFuse/

## 1 Introduction

Over the last few years, video action recognition has made rapid progress with the introduction of a number of large-scale video datasets (Carreira & Zisserman, 2017; Monfort et al., 2018; Goyal et al., 2017). Despite impressive results on commonly used benchmark datasets, efficiency remains a great challenge for many resource constrained applications due to the heavy computational burden of deep Convolutional Neural Network (CNN) models.

Motivated by the need of efficiency, extensive studies have been recently conducted that focus on either designing new lightweight architectures (e.g., R(2+1)D (Tran et al., 2018), S3D (Xie et al., 2018), channel-separated CNNs (Tran et al., 2019)) or selecting salient frames/clips conditioned on the input (Yeung et al., 2016; Wu et al., 2019b; Korbar et al., 2019; Gao et al., 2020). However, most of the existing approaches do not consider the fact that there exists redundancy in CNN features which can significantly save computation leading to more efficient action recognition. In particular, orthogonal to the design of compact models, the computational cost of a CNN model also has much to do with the redundancy of CNN features (Han et al., 2019). Furthermore, the amount of redundancy depends on the dynamics and type of events in the video: A set of still frames for a simple action (e.g. "Sleeping") will have a higher redundancy comparing to a fast-changed action with rich interaction and deformation (e.g. "Pulling two ends of something so that it gets stretched"). Thus, based on the input we could compute just a subset of features, while the rest of the channels can reuse history feature maps or even be skipped without losing any accuracy, resulting in large computational savings compared to computing all the features at a given CNN layer. Based on this intuition, we present a new perspective for efficient action recognition by adaptively deciding what channels to compute or reuse, on a per instance basis, for recognizing complex actions.

In this paper, we propose AdaFuse, an adaptive temporal fusion network that learns a decision policy to dynamically fuse channels from current and history feature maps for efficient action recognition. Specifically, our approach reuses history features when necessary (i.e., dynamically decides which channels to keep, reuse or skip per layer and per instance) with the goal of improving both recognition

---

*Email: mengyuethu@gmail.com. This work was done while Yue was an AI Resident at IBM Research.

accuracy and efficiency. As these decisions are discrete and non-differentiable, we rely on a Gumbel Softmax sampling approach (Jang et al., 2016) to learn the policy jointly with the network parameters through standard back-propagation, without resorting to complex reinforcement learning as in (Wu et al., 2019b; Fan et al., 2018; Yeung et al., 2016). We design the loss to achieve both competitive performance and resource efficiency required for action recognition. Extensive experiments on multiple benchmarks show that AdaFuse significantly reduces the computation without accuracy loss.

The main contributions of our work are as follows:

- We propose a novel approach that automatically determines which channels to keep, reuse or skip per layer and per target instance for efficient action recognition.
- Our approach is model-agnostic, which allows this to be served as a plugin operation for a wide range of 2D CNN-based action recognition architectures.
- The overall policy distribution can be seen as an indicator for the dataset characteristic, and the block-level distribution can bring potential guidance for future architecture designs.
- We conduct extensive experiments on four benchmark datasets (Something-Something V1 (Goyal et al., 2017), Something-Something V2 (Mahdisoltani et al., 2018), Jester (Materzynska et al., 2019) and Mini-Kinetics (Kay et al., 2017)) to demonstrate the superiority of our proposed approach over state-of-the-art methods.

## 2    RELATED WORK

**Action Recognition.** Much progress has been made in developing a variety of ways to recognize complex actions, by either applying 2D-CNNs (Karpathy et al., 2014; Wang et al., 2016; Fan et al., 2019) or 3D-CNNs (Tran et al., 2015; Carreira & Zisserman, 2017; Hara et al., 2018). Most successful architectures are usually based on the two-stream model (Simonyan & Zisserman, 2014), processing RGB frames and optical-flow in two separate CNNs with a late fusion in the upper layers (Karpathy et al., 2014) or further combining with other modalities (Asghari-Esfeden et al., 2020; Li et al., 2020a). Another popular approach for CNN-based action recognition is the use of 2D-CNN to extract frame-level features and then model the temporal causality using different aggregation modules such as temporal averaging in TSN (Wang et al., 2016), a bag of features scheme in TRN (Zhou et al., 2018), channel shifting in TSM (Lin et al., 2019), depthwise convolutions in TAM (Fan et al., 2019), non-local neural networks (Wang et al., 2018a), temporal enhancement and interaction module in TEINet (Liu et al., 2020), and LSTMs (Donahue et al., 2015). Many variants of 3D-CNNs such as C3D (Tran et al., 2015; Ji et al., 2013), I3D (Carreira & Zisserman, 2017) and ResNet3D (Hara et al., 2018), that use 3D convolutions to model space and time jointly, have also been introduced for action recognition. SlowFast (Feichtenhofer et al., 2018) employs two pathways to capture temporal information by processing a video at both slow and fast frame rates. Recently, STM (Jiang et al., 2019) proposes new channel-wise convolutional blocks to jointly capture spatio-temporal and motion information in consecutive frames. TEA (Li et al., 2020b) introduces a motion excitation module including multiple temporal aggregation modules to capture both short- and long-range temporal evolution in videos. Gate-Shift networks (Sudhakaran et al., 2020) use spatial gating for spatial-temporal decomposition of 3D kernels in Inception-based architectures.

While extensive studies have been conducted in the last few years, limited efforts have been made towards *efficient* action recognition (Wu et al., 2019b;a; Gao et al., 2020). Specifically, methods for efficient recognition focus on either designing new lightweight architectures that aim to reduce the complexity by decomposing the 3D convolution into 2D spatial convolution and 1D temporal convolution (e.g., R(2+1)D (Tran et al., 2018), S3D (Xie et al., 2018), channel-separated CNNs (Tran et al., 2019)) or selecting salient frames/clips conditioned on the input (Yeung et al., 2016; Wu et al., 2019b; Korbar et al., 2019; Gao et al., 2020). Our approach is most related to the latter which focuses on conditional computation and is agnostic to the network architecture used for recognizing actions. However, instead of focusing on data sampling, our approach dynamically fuses channels from current and history feature maps to reduce the computation. Furthermore, as feature maps can be redundant or noisy, we use a skipping operation to make it more efficient for action recognition.

**Conditional Computation.** Many conditional computation methods have been recently proposed with the goal of improving computational efficiency (Bengio et al., 2015; 2013; Veit & Belongie, 2018; Wang et al., 2018b; Graves, 2016; Meng et al., 2020; Pan et al., 2021). Several works have been

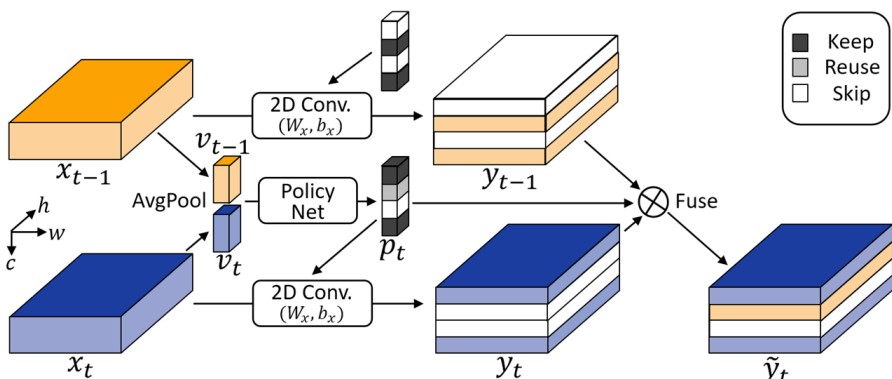

Figure 1: A conceptual view for adaptive temporal fusion. At time $t$, the 2D Conv layer computes for those "keep" channels (blue) in feature map $x_t$, and fuses the "reuse" channels (yellow) from the history feature map $y_{t-1}$. The downstream 2D Conv layer (not shown here) will process those "reuse" and "keep" channels in $\tilde{y}_t$. Best viewed in color.

proposed that add decision branches to different layers of CNNs to learn whether to exit the network for faster inference (Figurnov et al., 2017; McGill & Perona, 2017; Wu et al., 2020). BlockDrop (Wu et al., 2018) effectively reduces the inference time by learning to dynamically select which layers to execute per sample during inference. SpotTune (Guo et al., 2019) learns to adaptively route information through finetuned or pre-trained layers. Conditionally parameterized convolutions (Yang et al., 2019) or dynamic convolutions (Chen et al., 2019a; Verelst & Tuytelaars, 2019) have also been proposed to learn specialized convolutional kernels for each example to improve efficiency in image recognition. Our method is also related to recent works on dynamic channel pruning (Gao et al., 2018; Lin et al., 2017) that generate decisions to skip the computation for a subset of output channels. While GaterNet (Chen et al., 2019b) proposes a separate gating network to learn channel-wise binary gates for the backbone network, Channel gating network (Hua et al., 2019) identifies regions in the features that contribute less to the classification result, and skips the computation on a subset of the input channels for these ineffective regions. In contrast to the prior works that focus on only dropping unimportant channels, our proposed approach also reuses history features when necessary to make the network capable for strong temporal modelling.

## 3 METHODOLOGY

In this section, we first show the general approach using 2D-CNN for action recognition. Then we present the concept of adaptive temporal fusion and analyze its computation cost. Finally, we describe the end-to-end optimization and network specifications.

**Using 2D-CNN for Action Recognition.** One popular solution is to first generate frame-wise predictions and then utilize a consensus operation to get the final prediction (Wang et al., 2016). The network takes uniformly sampled $T$ frames $\{X_1...X_T\}$ and predicts the un-normalized class score:

$$P(X_1, ..., X_T; \Theta) = \mathcal{G}\left(\mathcal{F}(X_1; \Theta), \mathcal{F}(X_2; \Theta), ..., \mathcal{F}(X_T; \Theta)\right) \qquad (1)$$

where $\mathcal{F}(\cdot; \Theta)$ is the 2D-CNN with learnable parameters $\Theta$. The consensus function $\mathcal{G}$ reduces the frame-level predictions to a final prediction. One common practice for $\mathcal{G}$ is the averaging operation.

The major drawback is that this cannot capture the order of the frames. The network performs poorly on datasets that contain temporal-related labels (e.g. "turning left", "moving forward", etc). LSTM (Hochreiter & Schmidhuber, 1997) can also be used as $\mathcal{G}$ to get the final prediction (Donahue et al., 2015), but it cannot capture low-level features across the frames, as mentioned in Lin et al. (2019). A few works have been recently proposed to model temporal causality using a bag of features scheme in TRN (Zhou et al., 2018), channel shifting in TSM (Lin et al., 2019), depthwise convolutions in TAM (Fan et al., 2019). Different from these methods, in this work, we hypothesis that an *input-dependent* fusion of framewise features will be beneficial for temporal understanding and efficiency, as the amount of temporal information depends on the dynamics and the type of events in the video. Hence we propose adaptive temporal fusion for action recognition.

**Adaptive Temporal Fusion.** Consider a single 2D convolutional layer: $y_t = \phi(W_x * x_t + b_x)$, where $x_t \in \mathbb{R}^{c \times h \times w}$ denotes the input feature map at time step $t$ with $c$ channels and spatial dimension $h \times w$, and $y_t \in \mathbb{R}^{c' \times h' \times w'}$ is the output feature map. $W_x \in \mathbb{R}^{c' \times k \times k \times c}$ denotes the convolution filters (with kernel size $k \times k$) and $b_x \in \mathbb{R}^{c'}$ is the bias. We use "$*$" for convolution operation. $\phi(\cdot)$ is the combination of batchnorm and non-linear functions (e.g. ReLU (Nair & Hinton, 2010)).

We introduce a policy network consisting of two fully-connected layers and a ReLU function designed to adaptively select channels for keeping, reusing or skipping. As shown in Figure 1, at time $t$, we first generate feature vectors $v_{t-1}, v_t \in \mathbb{R}^c$ from history feature map $x_{t-1}$ and current feature map $x_t$ via global average pooling. Then the policy network predicts:

$$p_t = g(v_{t-1}, v_t; \Theta_g) \tag{2}$$

where $p_t \in \{0, 1, 2\}^{c'}$ is a channel-wise policy (choosing "keep", "reuse" or "skip") to generate the output feature map: if $p_t^i = 0$, the $i$-th channel of output feature map will be computed via the normal convolution; if $p_t^i = 1$, it will reuse the $i$-th channel of the feature map $y_{t-1}$ which has been already computed at time $t-1$; otherwise, the $i$-th channel will be just padded with zeros. Formally, this output feature map can be written as $\tilde{y}_t = f(y_{t-1}, y_t, p_t)$ where the $i$-th channel is:

$$\tilde{y}_t^i = \mathbb{1}\left[p_t^i = 0\right] \cdot y_t^i + \mathbb{1}\left[p_t^i = 1\right] \cdot y_{t-1}^i \tag{3}$$

here $\mathbb{1}[\cdot]$ is the indicator function. In Figure 1, the policy network instructs the convolution layer to only compute the first and fourth channels, reuses the second channel of the history feature and skips the third channel. Features from varied time steps are adaptively fused along the channel dimension.

Adaptive temporal fusion enables the 2D convolution to capture temporal information: its temporal perceptive field grows linearly to the depth of the layers, as more features from different time steps are fused when going deeper in the network. Our novel design can be seen as a general methodology for many state-of-the-art 2D-CNN approaches: if we discard "skip" and use a predefined fixed policy, then it becomes the online temporal fusion in Lin et al. (2019). If the policy only chooses from "skip" and "keep", then it becomes dynamic pruning methods (Gao et al., 2018; Hua et al., 2019). Our design is a generalized approach taking both temporal modelling and efficiency into consideration.

**Complexity Analysis.** To illustrate the efficiency of our framework, we compute the floating point operations (FLOPS), which is a hardware-independent metric and widely used in the field of efficient action recognition[1](Wu et al., 2019b; Gao et al., 2020; Meng et al., 2020; Fan et al., 2019). To compute saving from layers before and after the policy network, we add another convolution after $\tilde{y}_t$ with kernel $W_y \in \mathbb{R}^{c'' \times k' \times k' \times c'}$ and bias $b_y \in \mathbb{R}^{c''}$. The total FLOPS for each convolution will be:

$$\begin{cases} m_x = c' \cdot h' \cdot w' \cdot (k \cdot k \cdot c + 1) \\ m_y = c'' \cdot h'' \cdot w'' \cdot (k' \cdot k' \cdot c' + 1) \end{cases} \tag{4}$$

When the policy is applied, only those output channels used in time $t$ or going to be reused in time $t+1$ need to be computed in the first convolution layer, and only the channels not skipped in time $t$ count for input feature maps for the second convolution layer. Hence the overall FLOPS is:

$$M = \sum_{\tau=0}^{T-1} \left[ \underbrace{\frac{1}{c'} \sum_{i=0}^{c'-1} \mathbb{1}\left[p_\tau^i \cdot (p_{\tau+1}^i - 1) = 0\right] \cdot m_x}_{\text{FLOPS from the first conv at time } \tau} + \underbrace{\left(1 - \frac{1}{c'} \sum_{i=0}^{c'-1} \mathbb{1}(p_\tau^i = 2)\right) \cdot m_y}_{\text{FLOPS from the second conv at time } \tau} \right] \tag{5}$$

where the braces over the first sum read "Keep at $\tau$ or resue at $\tau + 1$" and over the second sum read "Skip at $\tau$".

Thus when the policy network skips more channels or reuses channels that are already computed in the previous time step, the FLOPS for those two convolution layers can be reduced proportionally.

**Loss functions.** We take the average of framewise predictions as the video prediction and minimize:

$$\mathcal{L} = \sum_{(x,y) \sim D_{train}} \left[ -y \log(P(x)) + \lambda \cdot \sum_{i=0}^{B-1} M_i \right] \tag{6}$$

---

[1]Latency is another important measure for efficiency, which can be reduced via CUDA optimization for sparse convolution (Verelst & Tuytelaars, 2019). We leave it for future research.

The first term is the cross entropy between one-hot encoded ground truth labels $y$ and predictions $P(x)$. The second term is the FLOPS measure for all the $B$ temporal fusion blocks in the network. In this way, our network is learned to achieve both accuracy and efficiency at a trade-off controlled by $\lambda$.

Discrete policies for "keep", "reuse" or "skip" shown in Eq. 3 and Eq. 5 make $\mathcal{L}$ non-differentiable hence hard to optimize. One common practice is to use a score function estimator (e.g. REINFORCE (Glynn, 1990; Williams, 1992)) to avoid backpropagating through categorical samplings, but the high variance of the estimator makes the training slow to converge (Wu et al., 2019a; Jang et al., 2016). As an alternative, we use Gumbel-Softmax Estimator to enable efficient end-to-end optimization.

**Training using Gumbel Softmax Estimator.** Specifically, the policy network first generates a logit $q \in \mathbb{R}^3$ for each channel in the output feature map and then we use Softmax to derive a normalized categorical distribution: $\pi = \{r_i | r_i = \frac{\exp(q_i)}{\exp(q_0) + \exp(q_1) + \exp(q_2)}\}$. With the Gumbel-Max trick, discrete samples from the distribution $\pi$ can be drawn as (Jang et al., 2016): $\hat{r} = \text{argmax}_i(\log r_i + G_i)$, where $G_i = -\log(-\log U_i)$ is a standard Gumbel distribution with i.i.d. $U_i$ sampled from a uniform distribution $\text{Unif}(0, 1)$. Since the argmax operator is not differentiable, the Gumbel Softmax distribution is used as a continuous approximation. In forward pass we represent the discrete sample $\hat{r}$ as a one-hot encoded vector and in back-propagation we relax it to a real-valued vector $R = \{R_0, R_1, R_2\}$ via Softmax as follows:

$$R_i = \frac{\exp\left((\log r_i + G_i)/\tau\right)}{\sum_{j=1}^{2} \exp\left((\log r_j + G_j)/\tau\right)} \tag{7}$$

where $\tau$ is a temperature factor controlling the "smooothness" of the distribution: $\lim_{\tau \to \infty} R$ converges to a uniform distribution and $\lim_{\tau \to 0} R$ becomes a one-hot vector. We set $\tau = 0.67$ during the training.

**Network Architectures and Notations.** Our adaptive temporal fusion module can be easily plugged into any existing 2D-CNN models. Specifically, we focus on BN-Inception (Ioffe & Szegedy, 2015), ResNet (He et al., 2016) and EfficientNet (Tan & Le, 2019). For Bn-Inception, we add a policy network between every two consecutive Inception modules. For ResNet/EfficientNet, we insert the policy network between the first and the second convolution layers in each "residual block"/"inverted residual block". We denote our model as $\text{AdaFuse}_{\text{Backbone}}^{\text{Method}}$, where the "Backbone" is chosen from {"R18"(ResNet18), "R50"(ResNet50), "Inc"(BN-Inception), "Eff"(EfficientNet)}, and the "Method" can be {"TSN", "TSM", "TSM+Last"}. More details can be found in the following section.

# 4 EXPERIMENTS

We first show AdaFuse can significantly improve the accuracy and efficiency of ResNet18, BN-Inception and EfficientNet, outperforming other baselines by a large margin on Something-V1. Then on all datasets, AdaFuse with ResNet18 / ResNet50 can consistently outperform corresponding base models. We further propose two instantiations using AdaFuse on TSM (Lin et al., 2019) to compare with state-of-the-art approaches on Something V1 & V2: $\text{AdaFuse}_{\text{R50}}^{\text{TSM}}$ can save over 40% FLOPS at a comparable classification score under same amount of computation budget, $\text{AdaFuse}_{\text{R50}}^{\text{TSM+Last}}$ outperforms state-of-the-art methods in accuracy. Finally, we perform comprehensive ablation studies and quantitative analysis to verify the effectiveness of our adaptive temporal fusion.

**Datasets.** We evaluate AdaFuse on Something-Something V1 (Goyal et al., 2017) & V2 (Mahdisoltani et al., 2018), Jester (Materzynska et al., 2019) and a subset of Kinetics (Kay et al., 2017). Something V1 (98k videos) & V2 (194k videos) are two large-scale datasets sharing 174 human action labels (e.g. pretend to pick something up). Jester (Materzynska et al., 2019) has 27 annotated classes for hand gestures, with 119k / 15k videos in training / validation set. Mini-Kinetics (assembled by Meng et al. (2020)) is a subset of full Kinetics dataset (Kay et al., 2017) containing 121k videos for training and 10k videos for testing across 200 action classes.

**Implementation details.** To make a fair comparison, we carefully follow the training procedure in Lin et al. (2019). We uniformly sample $T = 8$ frames from each video. The input dimension for the network is $224 \times 224$. Random scaling and cropping are used as data augmentation during training (and we further adopt random flipping for Mini-Kinetics). Center cropping is used during inference. All our networks are using ImageNet pretrained weights. We follow a step-wise learning rate scheduler with the initial learning rate as 0.002 and decay by 0.1 at epochs 20 & 40. To train

Table 1: Action Recognition Results on Something-Something-V1 Dataset. Our proposed method consistently outperforms all other baselines in both accuracy and efficiency.

| Method | Backbone | #Params | FLOPS | Top1 | Top5 |
|---|---|---|---|---|---|
| TSN (Wang et al., 2016) | ResNet18 | 11.2M | 14.6G | 14.8 | 38.0 |
| TSN (Wang et al., 2016) | BN-Inception | 10.4M | 16.4G | 17.6 | 43.5 |
| CGNet (Hua et al., 2019) | ResNet18 | 11.2M | 11.2G | 13.7 | 35.1 |
| Threshold | ResNet18 | 11.2M | 11.3G | 14.1 | 36.6 |
| Random | ResNet18 | 11.2M | **10.4G** | 27.5 | 54.2 |
| LSTM | ResNet18 | 11.7M | 14.7G | 28.4 | 56.3 |
| AdaFuse$_{R18}^{TSN}$ | ResNet18 | 15.6M | **10.3G** | **36.9** | **65.0** |
| AdaFuse$_{Inc}^{TSN}$ | BN-Inception | 14.5M | 12.1G | **38.5** | **67.8** |

Table 2: Action Recognition on Something-Something-V1 using EfficientNet architecture. AdaFuse$_{Eff-x}^{TSN}$ consistently outperforms all the EfficientNet baselines in both accuracy and efficiency.

| Method | Backbone | #Params | FLOPS | Top1 | Top5 |
|---|---|---|---|---|---|
| TSN | EfficientNet-b0 | 5.3M | **3.1G** | 18.0 | 44.9 |
| TSN | EfficientNet-b1 | 7.8M | 5.6G | 19.3 | 45.9 |
| TSN | EfficientNet-b2 | 9.2M | 8.0G | 18.8 | 46.0 |
| TSN | EfficientNet-b3 | 12.0M | 14.4G | 19.3 | 46.6 |
| AdaFuse$_{Eff-0}^{TSN}$ | EfficientNet-b0 | 9.3M | **2.8G** | 39.0 | 68.1 |
| AdaFuse$_{Eff-1}^{TSN}$ | EfficientNet-b1 | 12.4M | 4.9G | 40.3 | 69.2 |
| AdaFuse$_{Eff-2}^{TSN}$ | EfficientNet-b2 | 13.8M | 7.2G | **40.2** | **69.5** |
| AdaFuse$_{Eff-3}^{TSN}$ | EfficientNet-b3 | 16.6M | 12.9G | **40.7** | **69.7** |

our adaptive temporal fusion approach, we set the efficiency term $\lambda = 0.1$. We train all the models for 50 epochs with a batch-size of 64, where each experiment takes 12∼ 24 hours on 4 Tesla V100 GPUs. We report the number of parameters used in each method, and measure the averaged FLOPS and Top1/Top5 accuracy for all the samples from each testing dataset.

**Adaptive Temporal Fusion improves 2D CNN Performance.** On Something V1 dataset, we show AdaFuse 's improvement upon 2D CNNs by comparing with several baselines as follows:

- TSN (Wang et al., 2016): Simply average frame-level predictions as the video-level prediction.
- CGNet (Hua et al., 2019): A dynamic pruning method to reduce computation cost for CNNs.
- Threshold: We keep a fixed portion of channels base on their activation L1 norms and skip the channels in smaller norms. It serves as a baseline for efficient recognition.
- RANDOM: We use temporal fusion with a randomly sampled policy (instead of using learned policy distribution). The distribution is chosen to match the FLOPS of adaptive methods.
- LSTM: Update per-frame predictions by hidden states in LSTM and averages all predictions as the video-level prediction.

We implement all the methods using publicly available code and apply adaptive temporal fusion in TSN using ResNet18, BN-Inception and EfficientNet backbones, denoting them as AdaFuse$_{R18}^{TSN}$ AdaFuse$_{Inc}^{TSN}$ and AdaFuse$_{Eff-x}^{TSN}$ respectively ("x" stands for different scales of the EfficientNet backbones). As shown in Table 1, AdaFuse$_{R18}^{TSN}$ uses the similar FLOPS as those efficient methods ("CGNet" and "Threshold") but has a great improvement in classification accuracy Specifically, AdaFuse$_{R18}^{TSN}$ and AdaFuse$_{Inc}^{TSN}$ outperform corresponding TSN models by more than 20% in Top-1 accuracy, while using only 74% of FLOPS. Interestingly, comparing to TSN, even temporal fusion with a random policy can achieve an absolute gain of 12.7% in accuracy, which shows that temporal fusion can greatly improve the action recognition performance of 2D CNNs. Additionally equipped with the adaptive policy, AdaFuse$_{R18}^{TSN}$ can get 9.4% extra improvement in classification. LSTM is the most competitive baseline in terms of accuracy, while AdaFuse$_{R18}^{TSN}$ has an absolute gain of 8.5% in accuracy and uses only 70% of FLOPS. When using a more efficient architecture as shown in Table.2, our approach can still reduce 10% of the FLOPS while improving the accuracy by a large margin. To further validate AdaFuse being model-agnostic and robust, we conduct extensive experiments using ResNet18 and

Table 3: Comparison with TSN using ResNet-18/ResNet-50 backbones. AdaFuse consistently outperforms TSN by a large margin in accuracy while offering significant savings in FLOPs.

| Method | #Params | SomethingV1 | | SomethingV2 | | Jester | | Mini-Kinetics | |
|---|---|---|---|---|---|---|---|---|---|
| | | FLOPS | Top1 | FLOPS | Top1 | FLOPS | Top1 | FLOPS | Top1 |
| TSN$_{R18}$ (Wang et al., 2016) | 11.2M | 14.6G | 14.8 | 14.6G | 27.3 | 14.6G | 82.6 | 14.6G | 64.6 |
| LSTM$_{R18}$ | 11.7M | 14.7G | 28.4 | 14.7G | 40.3 | 14.7G | 93.5 | 14.7G | 67.2 |
| AdaFuse$_{R18}^{TSN}$ | 15.6M | **10.3G** | **36.9** | **11.1G** | **50.5** | **7.6G** | **93.7** | **11.8G** | **67.5** |
| TSN$_{R50}$ (Wang et al., 2016) | 23.6M | 32.9G | 18.7 | 32.9G | 32.1 | 32.9G | 82.6 | 32.9G | 72.1 |
| LSTM$_{R50}$ | 28.8M | 33.0G | 30.1 | 33.0G | 47.4 | 33.0G | 93.7 | 33.0G | 71.6 |
| AdaFuse$_{R50}^{TSN}$ | 37.8M | **22.1G** | **41.9** | **18.1G** | **56.8** | **16.1G** | **94.7** | **23.0G** | **72.3** |

Table 4: Comparison with the recent adaptive inference method AR-Net (Meng et al., 2020) on Something-Something-V1, Jester and Mini-Kinetics datasets. AdaFuse$_{R50}^{TSN}$ achieves a better accuracy with great savings in computation (FLOPS) and number of parameters.

| Method | #Params | SomethingV1 | | Jester | | Mini-Kinetics | |
|---|---|---|---|---|---|---|---|
| | | FLOPS | Top1 | FLOPS | Top1 | FLOPS | Top1 |
| AR-Net (Meng et al., 2020) | 63.0M | 41.4G | 18.9 | 21.2G | 87.8 | 32.0G | 71.7 |
| AdaFuse$_{R50}^{TSN}$ | **37.8M** | **22.1G** | **41.9** | **16.1G** | **94.7** | **23.0G** | **72.3** |

Table 5: Comparison with State-of-the-Art methods on Something-Something-V1 & V2 datasets. Our method has comparative accuracy with great savings in FLOPS.

| Method | Backbone | T | #Params | Something-V1 | | Something-V2 | |
|---|---|---|---|---|---|---|---|
| | | | | FLOPS | Top1 | FLOPS | Top1 |
| TSN (Wang et al., 2016) | BN-Inception | 8 | 10.7M | 16.0G | 19.5 | 16.0G | 33.4 |
| TSN (Wang et al., 2016) | ResNet50 | 8 | 24.3M | 33.2G | 19.7 | 33.2G | 27.8 |
| TRN$_{Multiscale}$ (Zhou et al., 2018) | BN-Inception | 8 | 18.3M | 16.0G | 34.4 | 16.0G | 48.8 |
| TRN$_{RGB+Flow}$ (Zhou et al., 2018) | BN-Inception | 8+8 | 36.6M | 32.0G | 42.0 | 32.0G | 55.5 |
| I3D (Carreira & Zisserman, 2017) | 3DResNet50 | 32×2 | 28.0M | 306G | 41.6 | - | - |
| I3D+GCN+NL (Wang & Gupta, 2018) | 3DResNet50 | 32×2 | 62.2M | 606G | 46.1 | - | - |
| ECO (Zolfaghari et al., 2018) | BNInc+3DRes18 | 8 | 47.5M | 32G | 39.6 | - | - |
| ECO$_{En}$Lite (Zolfaghari et al., 2018) | BNInc+3DRes18 | 92 | 150M | 267G | **46.4** | - | - |
| TSM (Lin et al., 2019) | ResNet50 | 8 | 24.3M | 33.2G | 45.6 | 33.2G | **59.1** |
| AdaFuse$_{Inc}^{TSN}$ | BN-Inception | 8 | 14.5M | 12.1G | 38.5 | 12.5G | 53.4 |
| AdaFuse$_{R50}^{TSN}$ | ResNet50 | 8 | 37.7M | 22.1G | 41.9 | 18.1G | 56.8 |
| AdaFuse$_{R50}^{TSM}$ | ResNet50 | 8 | 37.7M | 19.1G | 44.9 | 19.5G | 58.3 |
| AdaFuse$_{R50}^{TSM+Last}$ | ResNet50 | 8 | 39.1M | 31.5G | **46.8** | 31.3G | 59.8 |

ResNet50 backbones on Something V1 & V2, Jester and Mini-Kinetics. As shown in Table 3, AdaFuse$_{R18}^{TSN}$ and AdaFuse$_{R50}^{TSN}$ consistently outperform their baseline TSN and LSTM models with a 35% saving in FLOPS on average. Our approach harvests large gains in accuracy and efficiency on temporal-rich datasets like Something V1 & V2 and Jester. When comes to Mini-Kinetics, AdaFuse can still achieve a better accuracy with 20%~33% computation reduction.

**Comparison with Adaptive Inference Method.** We compare our approach with AR-Net (Meng et al., 2020), which adaptively chooses frame resolutions for efficient inference. As shown in Table 4, on Something V1, Jester and Mini-Kinetics, we achieve a better accuracy-efficiency trade-off than AR-Net while using 40% less parameters. On temporal-rich dataset like Something-V1, our approach attains the largest improvement, which shows AdaFuse$_{R50}^{TSN}$'s capability for strong temporal modelling.

**Comparison with State-of-the-Art Methods.** We apply adaptive temporal fusion with different backbones (ResNet50 (He et al., 2016), BN-Inception (Ioffe & Szegedy, 2015)) and designs (TSN (Wang et al., 2016), TSM (Lin et al., 2019)) and compare with State-of-the-Art methods on Something V1 & V2. As shown in Table 5, using BN-Inception as backbone, AdaFuse$_{Inc}^{TSN}$ is 4% better than "TRN$_{Multiscale}$" (Zhou et al., 2018) in accuracy, using only 75% of the FLOPS. AdaFuse$_{R50}^{TSN}$ with ResNet50 can even outperform 3D CNN method "I3D" (Carreira & Zisserman, 2017) and hybrid 2D/3D CNN method "ECO" (Zolfaghari et al., 2018) with much less FLOPS.

As for adaptive temporal fusion on "TSM" (Lin et al., 2019), AdaFuse$_{R50}^{TSM}$ achieves more than 40% savings in computation but at 1% loss in accuracy (Table 5). We believe this is because TSM uses temporal shift operation, which can be seen as a variant of temporal fusion. Too much temporal fusion could cause performance degradation due to a worse spatial modelling capability. As a remedy, we just adopt adaptive temporal fusion in the last block in TSM to capture high-level semantics (more intuition can be found later in our visualization experiments) and denote it as AdaFuse$_{R50}^{TSM+Last}$. On Something V1 & V2 datasets, AdaFuse$_{R50}^{TSM+Last}$ outperforms TSM and all other state-of-the-art methods in accuracy with a 5% saving in FLOPS comparing to TSM. From our experiments, we observe that the performance of adaptive temporal fusion depends on the position of shift modules in TSM and optimizing the position of such modules through additional regularization could help us not only to achieve better accuracy but also to lower the number of parameters. We leave this as an interesting future work.

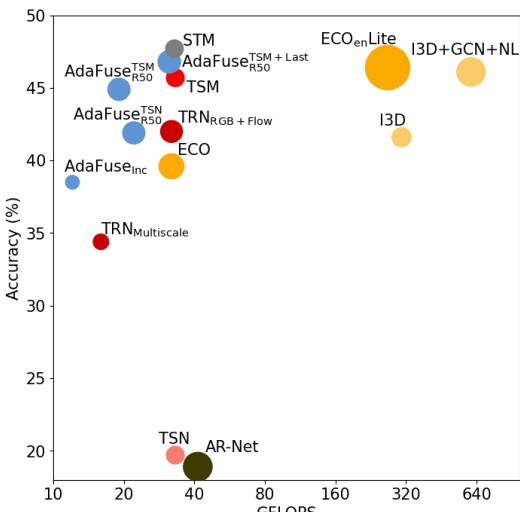

Figure 2: FLOPS vs Accuracy on Something-V1 Dataset. The diameter of each data point is proportional to the total number of parameters. AdaFuse (blue points) achieves the best trade-off at a comparable model size to 2D CNN approaches.

We depict the accuracy, computation cost and model sizes in Figure 2. All the results are computed from Something V1 validation set. The graph shows GFLOPS / accuracy on x / y-axis and the diameter of each data point is proportional to the number of model parameters. AdaFuse (blue points) owns the best trade-off for accuracy and efficiency at a comparable model size to other 2D CNN approaches. Once again it shows AdaFuse is an effective and efficient design for action recognition.

**Policy Visualizations.** Figure 3 shows overall policy ("Skip", "Reuse" and "Keep") differences across all datasets. We focus on the quotient of "Reuse / Keep" as it indicates the mixture ratio for feature fusion. The quotients on Something V1&V2 and Jester datasets are very high (0.694, 0.741 and 0.574 respectively) when comparing to Mini-Kinetics (0.232). This is probably because the first three datasets contain more temporal relationship than Kinetics. Moreover, Jester has the highest percentage in skipping which indicates many actions in this dataset can be correctly recognized with few channels: Training on Jester is more biased towards optimizing for efficiency as the accuracy loss is very low. Distinctive policy patterns show different characteristics of datasets, which conveys a potential of our proposed approach to be served as a "dataset inspector".

Figure 4 shows a more fine-grained policy distribution on Something V2. We plot the policy usage in each residual block inside the ResNet50 architecture (shown in light red/orange/blue) and use 3rd-order polynomials to estimate the trend of each policy (shown in black dash curves). To further study the time-sensitiveness of the policies, we calculate the number of channels where the policies stay unchanged across the frames in one video (shown in dark red/orange/blue). We find earlier layers tend to skip more and reuse/keep less, and vice versa. The first several convolution blocks normally capture low-level feature maps in large spatial sizes, so the "information density" on channel dimension should be less which results in more redundancy across channels. Later blocks often capture high-level semantics and the feature maps are smaller in spatial dimensions, so the "semantic density" could be higher and less channels will be skipped. In addition, low-level features change faster across the frames (shades, lighting intensity) whereas high-level semantics change slowly across the frames (e.g. "kicking soccer"), that's why more features can be reused in later layers to avoid computing the same semantic again. As for the time-sensitiveness, earlier layers tend to be less sensitive and vice versa. We find that "reuse" is the most time-sensitive policy, as "Reuse (Instance)" ratio is very low, which again shows the functioning of adaptive temporal fusion. We believe these findings will provide insights to future designs of effective temporal fusions.

**How does the adaptive policy affect the performance?** We consider AdaFuse$_{R18}^{TSN}$ on Something V1 dataset and break down by using "skip", 'reuse' and adaptive (Ada.) policy learning. As shown

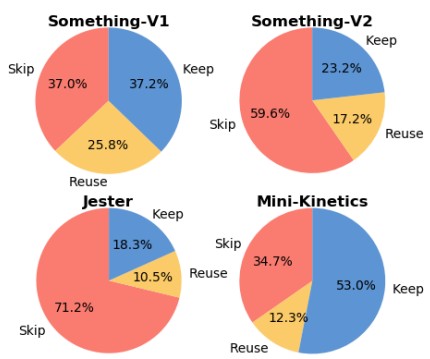

Figure 3: Dataset-specific policy distribution.

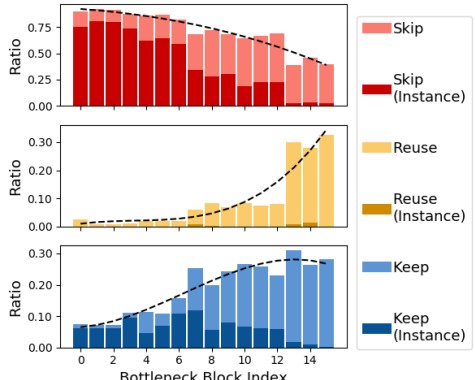

Figure 4: Policy distribution and trends for each residual block on Something-V2 dataset.

Table 6: Effect of different policies (using AdaFuse$_{R18}^{TSN}$) on Something V1 dataset.

| Method | Skip | Reuse | Ada. | FLOPS | Top1 |
|---|---|---|---|---|---|
| TSN | ✗ | ✗ | ✗ | 14.6G | 14.8 |
| Ada. Skip | ✔ | ✗ | ✔ | **6.6G** | 9.5 |
| Ada. Reuse | ✗ | ✔ | ✔ | 13.8G | 36.3 |
| Random | ✔ | ✔ | ✗ | 10.4G | 27.5 |
| AdaFuse$_{R18}^{TSN}$ | ✔ | ✔ | ✔ | **10.3G** | **36.9** |

Table 7: Effect of hidden sizes and efficient weights on the performance of AdaFuse$_{R50}^{TSM+Last}$ on SthV2.

| #Hidden Units | $\lambda$ | #Params | FLOPS | Top1 | Skip | Reuse |
|---|---|---|---|---|---|---|
| 1024 | 0.050 | 39.1M | 31.53G | 59.71 | 13% | 14% |
| 1024 | 0.075 | 39.1M | 31.29G | 59.75 | 15% | 13% |
| 1024 | 0.100 | 39.1M | **31.04G** | 59.40 | 18% | 12% |
| 2048 | 0.100 | 54.3M | **30.97G** | **59.96** | 21% | 10% |
| 4096 | 0.100 | 84.7M | 31.04G | **60.00** | 25% | 8% |

in Table 6, "Ada. Skip" saves 55% of FLOPS comparing to TSN but at a great degradation in accuracy. This shows naively skipping channels won't give a better classification performance. "Ada. Reuse" approach brings 21.5% absolute gain in accuracy, which shows the importance of temporal fusion. However, it fails to save much FLOPS due to the absence of skipping operation. Combining "Keep" with both "Skip" and "Reuse" via just a random policy is already achieving a better trade-off comparing to TSN, and by using adaptive learning approach, AdaFuse$_{R18}^{TSN}$ reaches the highest accuracy with the second-best efficiency. In summary, the "Skip" operation contributes the most to the computation efficiency, the "Reuse" operation boosts the classification accuracy, while the adaptive policy adds the chemistry to the whole system and achieves the best performance.

**How to achieve a better performance?** Here we investigate different settings to improve the performance of AdaFuse$_{R50}^{TSM+Last}$ on Something V2 dataset. As shown in Table 7, increasing $\lambda$ will obtain a better efficiency but might result in accuracy degradation. Enlarging the number of hidden units for the policy network can get a better overall performance: as we increase the size from 1024 to 4096, the accuracy keeps increasing. When the policy network grows larger, it learns to skip more to reduce computations and to reuse history features wisely for recognition. But notice that the model size grows almost linearly to hidden layer sizes, which leads to a considerable overhead to the FLOPS computation. As a compromise, we only choose $\lambda = 0.75$ and hidden size 1024 for AdaFuse$_{R50}^{TSM+Last}$. We leave the design for a more advanced and delicate policy module for future works.

**Runtime/Hardware.** Sparse convolutional kernels are often less efficient on current hardwares, e.g., GPUs. However, we strongly believe that it is important to explore models for efficient video action recognition which might guide the direction of new hardware development in the years to come. Furthermore, we also expect wall-clock time speed-up in the inference stage via efficient CUDA implementation, which we anticipate will be developed.

## 5 CONCLUSIONS

We have shown the effectiveness of adaptive temporal fusion for efficient video recognition. Comprehensive experiments on four challenging and diverse datasets present a broad spectrum of accuracy-efficiency models. Our approach is model-agnostic, which allows it to be served as a plugin operation for a wide range of architectures for video recognition tasks.

**Acknowledgements.** This work is supported by the Intelligence Advanced Research Projects Activity (IARPA) via DOI/IBC contract number D17PC00341. The U.S. Government is authorized to reproduce and distribute reprints for Governmental purposes notwithstanding any copyright annotation thereon. This work is also partly supported by the MIT-IBM Watson AI Lab.

**Disclaimer.** The views and conclusions contained herein are those of the authors and should not be interpreted as necessarily representing the official policies or endorsements, either expressed or implied, of IARPA, DOI/IBC, or the U.S. Government.

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
