# OpenReview forum: "AdaFuse: Adaptive Temporal Fusion Network for Efficient Action Recognition"
_ICLR.cc/2021/Conference — ICLR 2021 Poster_

### Official Review · AnonReviewer1 · 2020-10-20
**An Adaptive Temporal Fusion (AdaFuse) network has been proposed to balance efficiency and accuracy in video action recognition**

**Rating:** 6
**Confidence:** 4

**Review:**

In this work, the authors introduce an AdaFuse network for efficiency action recognition in videos. Specifically, they design a policy net to decide which channels should be kept, reused or skipped, according to the input features of two adjacent frames.

Strength

1 The paper is written well, and the organization is OK

2 The idea of adaptive temporal fusion is somehow novel and interesting

Weakness

1 How to save computation. I understand the general idea of saving computation, if some channels are reused or skipped. However, in the training phase, the policy net would produce the real-value vector by Eq. (7), instead of the one-hot vector. In other words, the 'keep' entry for each channel is always used during training. Then, I guess computation saving is not claimed for training. It is for testing, right? How to do test? The policy net produces the real-value vector and then you make it as one-hot vector for saving computation?

2 Missing SOTA. Compared with this paper, many recent approaches can achieve a competitive computation with better accuracy. It significantly reduces the potential value of this paper.

*Jiang et al., STM: SpatioTemporal and Motion Encoding for Action Recognition, ICCV 2019

*Li et al., TEA: Temporal Excitation and Aggregation for Action Recognition, CVPR 2020

*Sudhakaran et al., Gate-Shift Networks for Video Action Recognition, CVPR2020

*Liu et al., TEINet: Towards an efficient architecture for video recognition, AAAI 2020

3 Please correct the abstract. The experiments are performed on mini-Kinetics, rather than Kinetics. I indeed suggest that, it would be better to perform the proposed method on Kinetics to further show the effectiveness.

---

> ### Author Response · Authors · 2020-11-22
> **Response to AnonReviewer1 (Part 1)**
>
> Thanks for the thoughtful reviews and constructive suggestions.
>
> **(a) Computation saving:** Yes, you are right. Our method is mainly for efficient inference. Specifically, at the test stage, our policy network will cast the real-value vector to one-hot encoding and then chooses the corresponding channels to keep or reuse or skip. And when we use “reuse” or “skip”, we are saving the computation from those convolution operations.
>
> **(b) Missing SOTA:** Thanks for pointing to the recent references--we have cited them in our revised version. In this paper, our main goal is to reduce the computation cost of existing architectures (improve efficiency) while achieving a better classification accuracy - and our algorithm is also model-agnostic, making it easy to use as a plugin operation for other network architectures. STM [1] does not aim for improving the efficiency of existing networks - it proposes a new conv block using extra operations to capture the motion information in consecutive frames and improves the performance by an extra computation cost. Similarly, TEA [2] and TEINet [4] achieve higher accuracy with extra computations (from Motion Excitation & Temporal Aggregation and Motion Enhanced & Temporal Interaction Modules, respectively). Moreover, both TEA and TEINet use the same amount of computation for all the videos unlike the problem of adaptive inference that we consider in this paper. Gate-Shift Network [3] is only implemented on Inception-based architectures, where ours can be model-agnostic and can potentially combine with any 2D CNN-based architectures and can bring a better accuracy-efficiency trade-off compared to baseline methods.
>
> Since our approach is closely related to adaptive video inference models, as suggested by the reviewers, we compare with the state-of-the-art adaptive method, AR-Net (Meng et al. 2020, ECCV 2020) [5] on multiple datasets and observe that AdaFuse significantly outperforms AR-Net in both accuracy and efficiency with 40% less parameters (Table 4 in the revised paper). While AR-Net fails to capture the important temporal information present in datasets like Something-Something-V1, our approach adaptively reuses history features when necessary to make the network capable for strong temporal modelling. We also compare with a temporal fusion method, namely LSTM baseline and find that AdaFuse consistently outperforms LSTM models with a 35% savings on FLOPS on average (Table 3 in the revised paper). We summarize the new results as follows.
>
> $$\\begin{array} {|lrrrr|}
> \\hline \\text{Method (ACC/FLOPS)} & \\text{Params} & \\text{Something V1} & \\text{Jester} & \\text{Mini-Kinetics} \\\\
> \\hline \\text{AR-Net (Meng et al. 2020)} & \\text{63.0M} & \\text{18.9/41.4G} &  \\text{87.8/21.2G} & \\text{71.7/32.0G} \\\\
> \\text{AdaFuse (TSN-ResNet50)}& \\text{37.8M} & \\text{41.9/22.1G} & \\text{94.7/23.0G}  & \\text{72.3/23.0G} \\\\
> \\hline
>  \\end{array}$$
>
> $$\\begin{array} {|lrrrr|}
> \\hline \\text{Method (ACC/FLOPS)} & \\text{Something V1} & \\text{Something V2} & \\text{Jester} & \\text{Mini-Kinetics} \\\\
> \\hline \\text{LSTM (TSN-ResNet18)} & \\text{28.4/14.7G} & \\text{40.3/14.7G} &  \\text{93.5/14.7G} & \\text{67.2/14.7G} \\\\
> \\text{AdaFuse (TSN-ResNet18)} & \\text{36.9/10.3G} & \\text{50.5/11.2G} & \\text{93.7/7.6G}  & \\text{67.5/11.8G} \\\\
> \\hline
> \\text{LSTM (TSN-ResNet50)} & \\text{30.1/33.0G} & \\text{47.4/33.0G} & \\text{93.7/33.0G}  & \\text{71.6/33.0G} \\\\
> \\text{AdaFuse (TSN-ResNet50)} & \\text{41.9/22.1G} & \\text{56.8/18.1G} & \\text{94.7/16.1G}  & \\text{72.3/23.0G} \\\\
> \\hline
>  \\end{array}$$

---

> > ### Author Response · Authors · 2020-11-22
> > **Response to AnonReviewer1 (Part 2)**
> >
> > Furthemore, during the rebuttal period, we tried AdaFuse on top of EfficientNet - a more recent network architecture to verify the effectiveness of the proposed method. We kept almost all the hyperparameters the same as what we get from the ResNet experiments and the results on SomethingV1 are shown as followed:
> >
> > $$\\begin{array} {|lrrrr|}
> > \\hline \\text{Method} & \\text{\\#Params} & \\text{FLOPS} & \\text{Top1} & \\text{Top5} \\\\
> > \\hline
> > \\text{TSN (Eff-b0)} & \\text{5.3M} & \\text{3.1G} & \\text{18.0} & \\text{44.9} \\\\
> > \\text{AdaFuse (TSN-Eff-b0)} & \\text{9.3M} & \\text{2.8G} & \\text{39.0} & \\text{68.1} \\\\
> > \\hline
> > \\text{TSN (Eff-b1)} & \\text{7.8M} & \\text{5.6G} & \\text{19.3}  & \\text{45.9} \\\\
> > \\text{AdaFuse (TSN-Eff-b1)} & \\text{12.4M} & \\text{4.9G} & \\text{40.3}  & \\text{69.2} \\\\
> > \\hline
> > \\text{TSN (Eff-b2)} & \\text{9.2M} & \\text{8.0G} & \\text{18.8} & \\text{46.0} \\\\
> > \\text{AdaFuse (TSN-Eff-b2)} & \\text{13.8M} & \\text{7.2G} & \\text{40.2} & \\text{69.5} \\\\
> > \\hline
> > \\text{TSN (Eff-b3)} & \\text{12.0M} & \\text{14.4G} & \\text{19.3}  & \\text{46.6} \\\\
> > \\text{AdaFuse (TSN-Eff-b3)} & \\text{16.6M} & \\text{12.9G} & \\text{40.7}  & \\text{69.7} \\\\
> > \\hline
> >  \\end{array}$$
> >
> >
> > As can be seen from the above results, AdaFuse can still bring improvement in accuracy and efficiency even on more advanced and more efficient network architectures, which shows its power. In summary, our initial experiments including these new comparisons well demonstrate the efficiency of the proposed method over existing alternatives in efficient action recognition. Besides, not only can our method improve the performance of existing network architectures, but its “adaptive” property can also provide the interpretability to the network designs and the dataset used, as discussed in Section 4 of the main paper. We have updated all these new results in the revised version (see Table 2, Table 3 and Table 4).
> >
> > **(c) Abstract:** Thanks for pointing it out and we have fixed this in the revised paper. Since the full Kinetics dataset is quite large and the original version is no longer available from the official site (about ~15% videos are missing), we use the Mini-Kinetics dataset in our experiments. Besides Mini-Kinetics, we have also used three other temporal rich datasets, namely Something-Something V1/V2 and Jester to show the effectiveness of our method over existing methods.
> >
> > **References:**
> >
> > [1] Jiang et al., STM: SpatioTemporal and Motion Encoding for Action Recognition, ICCV 2019
> >
> > [2] Li et al., TEA: Temporal Excitation and Aggregation for Action Recognition, CVPR 2020
> >
> > [3] Sudhakaran et al., Gate-Shift Networks for Video Action Recognition, CVPR2020
> >
> > [4] Liu et al., TEINet: Towards an efficient architecture for video recognition, AAAI 2020
> >
> > [5] Meng et al., AR-Net: Adaptive Frame Resolution for Efficient Action Recognition, ECCV 2020

---

### Official Review · AnonReviewer4 · 2020-10-28
**Weakness in the experiment to support the effectiveness of the proposed method and unclarity in some of the important details**

**Rating:** 5
**Confidence:** 4

**Review:**

#### General
This paper proposes an adaptive temporal fusion network called AdaFuse for action recognition, which adaptively removes temporal redundancy and reuses past features for accuracy and efficiency.
I listed the Pros and Cons I found in the paper below as well as some questions to clarify some of the details.

#### Pros
1. The idea of learning a decision policy to dynamically determine whether channel-wise features at time $t$ are calculated normally, reused from $t-1$, or skipped, is interesting and reasonable.
1. The experimental results show that the proposed method achieves good accuracy with reasonable computational budget.
1. The ablation study in Table 4 reveals that the performance is greatly affected by the policy and it is important to fuse the futures from different frames to captures the temporal dependencies.

#### Cons
1. The propsoed method is not compared with some of the recent methods such as [1-3] ([4] is optional because the publication date is very close to the ICLR 2021 submission deadline). Especially for Jester and Mini-Kinetics dataset, the proposed method is compared with only TSN, which is old and weak as baseline as it does not incorporate the temporal information.
1. In Table 3, it seems that the proposed method achieves good accuracy, but I am afraid that it is just because of the strong base network, TSM. Merely adding AdaFuse to TSM indeed saves some computation but degrades the performance as described in the paper. The proposed remedy indeed slightly improves the accuracy but it requires much more parameters compared to the vanilla TSM. Overall, I find it benefitial to use the proposed method on top of simple base networks such as TSN, but the benefit of using the proposed method on top of strong base networks such as TSM may be marginal. Combined with the point 1 above, I am not well convinced of the effectiveness of the proposed method.
1. Some of the important details are not clear. I would appreciate if the authors could answer the questions I listed below.

#### Questions
1. Is it necessary to use Gumbel softmax? I think there are two kinds of tricks involved in Gumbel softmax. One is a trick for sampling from a categorical distribution, and the other is a trick for making the opperation differentiable. In my understanding, which may be wrong, the required characteristic for the present method is the latter one, and the sampling from the categorical distribution is not necessarily required. In this case, I think simply using $q$ instead of $\log{r} + G$ in equation (7) is enough.
1. Related to the point above, please clarify the type of output (hard or soft) of the policy net. The sentence after equation (2) says the output is integer values (0, 1, or 2), while the sentence before equation (7) says it is a real-valued vector.
1. Suppose $p_t^i = 1$ (reuse) and $p_{t-1}^i = 1$ (reuse again). In this case, is $y_t^i$ copied from $y_{t-2}^i$ ? Or is the feature map of $i$-th channel at time $t-1$ calculated on the fly for "reusing" at time $t$? In other words, if the policies for a channel is "reuse" $n$ consecutive times, does the method take the feature from $n$ frames before?


#### Other comments
1. Figure 1 may be incorrect or misleading. I think $p_t$, the output of the policy net, should go to the 2D Conv. block. Otherwise the block never knows which channel to compute at time $t$ and which channel to reuse or skip.

[1] Sudhakaran+, Gate-Shift Networks for Video Action Recognition, CVPR 2020
[2] Martinez+, Action recognition with spatial-temporal discriminative filter banks, ICCV 2019
[3] Jiang+, STM: SpatioTemporal and Motion Encoding for Action RecognitionSTM: SpatioTemporal and Motion Encoding for Action Recognition, ICCV 2019
[4] Kwon+, MotionSqueeze: Neural Motion Feature Learning for Video Understanding, ECCV 2020

---

> ### Author Response · Authors · 2020-11-22
> **Response to AnonReviewer4 (Part 1)**
>
> Thanks for the thoughtful reviews and constructive suggestions. We have incorporated all your suggestions including new comparisons with recent methods in the revised version.
>
> **(a) Comparison with recent methods:** Thanks for pointing to the recent references--we have cited them in our revised version. In this paper, our main goal is to reduce the computation cost of existing architectures (improve efficiency) while achieving a better classification accuracy - and our algorithm is also model-agnostic, making it easy to use as a plugin operation for other network architectures. Works in [1-3] are not about efficient action recognition and they usually focus on improving the classification accuracy by adding extra operations, which brings more FLOPS. The work in [4] focuses on replacing external and heavy computation of optical flows with internal learning of motion features. Moreover, most of them use the same amount of computation for all the videos unlike the problem of adaptive inference we consider in this paper. Besides, [1] is only implemented on Inception-based architectures, where ours is model-agnostic and can potentially combine with any sort of 2D CNN-based architectures and can bring a better accuracy-efficiency trade-off in video action recognition.
>
> Extensive experiments on four datasets with different backbone networks show that our adaptive temporal fusion approach is able to achieve good accuracy with reasonable computational budget. Since our approach is closely related to adaptive video inference models, as suggested by the reviewers, we compare with the recent adaptive method, AR-Net (Meng et al. 2020, ECCV 2020) on multiple datasets and observe that AdaFuse significantly outperforms AR-Net in both accuracy and efficiency with 40% less parameters (Table 4 in the revised paper). While AR-Net fails to capture the important temporal information present in datasets like Something-Something-V1, our approach adaptively reuses history features when necessary to make the network capable for strong temporal modelling. We also compare with a temporal fusion method, namely LSTM baseline and find that AdaFuse consistently outperforms LSTM models with a 35% savings on FLOPS on average (Table 3 in the revised paper). We summarize the results as follows.
>
> $$\\begin{array} {|lrrrr|}
> \\hline \\text{Method (ACC/FLOPS)} & \\text{Params} & \\text{Something V1} & \\text{Jester} & \\text{Mini-Kinetics} \\\\
> \\hline \\text{AR-Net (Meng et al. 2020)} & \\text{63.0M} & \\text{18.9/41.4G} &  \\text{87.8/21.2G} & \\text{71.7/32.0G} \\\\
> \\text{AdaFuse (TSN-ResNet50)} & \\text{37.8M} & \\text{41.9/22.1G} & \\text{94.7/23.0G}  & \\text{72.3/23.0G} \\\\
> \\hline
>  \\end{array}$$
>
> $$\\begin{array} {|lrrrr|}
> \\hline \\text{Method (ACC/FLOPS)} & \\text{Something V1} & \\text{Something V2} & \\text{Jester} & \\text{Mini-Kinetics} \\\\
> \\hline \\text{LSTM (TSN-ResNet18)} & \\text{28.4/14.7G} & \\text{40.3/14.7G} &  \\text{93.5/14.7G} & \\text{67.2/14.7G} \\\\
> \\text{AdaFuse (TSN-ResNet18)} & \\text{36.9/10.3G} & \\text{50.5/11.2G} & \\text{93.7/7.6G}  & \\text{67.5/11.8G} \\\\
> \\hline
> \\text{LSTM (TSN-ResNet50)} & \\text{30.1/33.0G} & \\text{47.4/33.0G} & \\text{93.7/33.0G}  & \\text{71.6/33.0G} \\\\
> \\text{AdaFuse (TSN-ResNet50)} & \\text{41.9/22.1G} & \\text{56.8/18.1G} & \\text{94.7/16.1G}  & \\text{72.3/23.0G} \\\\
> \\hline
>  \\end{array}$$
>
>
> Furthemore, during the rebuttal period, we tried AdaFuse on top of EfficientNet - a more recent network architecture to verify the effectiveness of the proposed method. We kept almost all the hyperparameters the same as what we get from the ResNet experiments and the results on SomethingV1 are shown as followed:
>
> $$\\begin{array} {|lrrrr|}
> \\hline \\text{Method} & \\text{\\#Params} & \\text{FLOPS} & \\text{Top1} & \\text{Top5} \\\\
> \\hline
> \\text{TSN (Eff-b0)} & \\text{5.3M} & \\text{3.1G} & \\text{18.0} & \\text{44.9} \\\\
> \\text{AdaFuse (TSN-Eff-b0)} & \\text{9.3M} & \\text{2.8G} & \\text{39.0} & \\text{68.1} \\\\
> \\hline
> \\text{TSN (Eff-b1)} & \\text{7.8M} & \\text{5.6G} & \\text{19.3}  & \\text{45.9} \\\\
> \\text{AdaFuse (TSN-Eff-b1)} & \\text{12.4M} & \\text{4.9G} & \\text{40.3}  & \\text{69.2} \\\\
> \\hline
> \\text{TSN (Eff-b2)} & \\text{9.2M} & \\text{8.0G} & \\text{18.8} & \\text{46.0} \\\\
> \\text{AdaFuse (TSN-Eff-b2)} & \\text{13.8M} & \\text{7.2G} & \\text{40.2} & \\text{69.5} \\\\
> \\hline
> \\text{TSN (Eff-b3)} & \\text{12.0M} & \\text{14.4G} & \\text{19.3}  & \\text{46.6} \\\\
> \\text{AdaFuse (TSN-Eff-b3)} & \\text{16.6M} & \\text{12.9G} & \\text{40.7}  & \\text{69.7} \\\\
> \\hline
>  \\end{array}$$
>
> As we can see, AdaFuse can still bring improvement in accuracy and efficiency even on more advanced and more efficient network architectures, which shows its power.

---

> > ### Author Response · Authors · 2020-11-22
> > **Response to AnonReviewer4 (Part 2)**
> >
> > Furthermore, while using adaptive temporal fusion in all the blocks of “TSM” (Lin et al., 2019), AdaFuse achieves more than 40% savings in computing budget but only at 0.7%-0.8% loss in accuracy. As TSM already uses temporal shift operation, we observe too much temporal fusion causes performance degradation due to a worse spatial modelling capability, consistent with the findings in (Lin et al., 2019). On the other hand, using adaptive temporal fusion only in the last block outperforms TSM with a 5% saving in FLOPS (Table 5) on both datasets. From our experiments, we observe that the performance of adaptive temporal fusion depends on the position of shift modules in TSM and optimizing the position of such modules through additional regularization could help us not only to achieve better accuracy but also to lower the number of parameters. We leave this as an interesting future work. In summary, our initial experiments including these new comparisons well demonstrate the efficiency of the proposed method over existing alternatives in efficient action recognition. Besides, not only can our method improve the performance of existing network architectures, but its “adaptive” property can also provide the interpretability to the network designs and the dataset used, as discussed in Section 4 of the paper. We have updated all these new results in the revised version (see Table 2, Table 3 and Table 4).
> >
> > **(b) Necessity of Gumbel Softmax:** We do need to use the sampling process from Gumbel Softmax, because we require discrete values to “select” some of the channels and “omit” other channels to save computation, whereas those logits q cannot achieve this - using those logits are more like a soft attention (continuous values), where we combine everything in a weighted sum therefore it doesn’t reduce any FLOPS. In particular, since we need the hard attention (discrete values) to skip channels, we adopt Gumbel Softmax sampling in our approach. Gumbel Softmax is a correct way of sampling from a discrete distribution which is parametrized by an unconstrained vector of numbers (Jang et al. 2016).
> >
> > **(c) Output type of the policy net:** The output type of the policy network is “hard” in the forward pass, and soft in the backward propagation. The motivation of this design is based on the Straight-Through Estimator [6]. In this way we can select the channels using the hard output, while learning from the soft values. The implementation can be found in the PyTorch’s official implementation of the gumbel_softmax function defined in:  [https://pytorch.org/docs/stable/_modules/torch/nn/functional.html#gumbel_softmax](https://pytorch.org/docs/stable/_modules/torch/nn/functional.html#gumbel_softmax). We have updated it in the revised paper.
> >
> > **(d) Reusing multiple times:**  Our method does not take the feature from n prior frames. Our definition of “reuse” is defined as “use the feature from the previous timestep’s convolutional feature map”, i.e., we only consider the feature map from the last time stamp (t-1) as we observe that merging features from frames far away (e.g. features from the current frame and features from n frames before) did not help in the convergence of the network, leading to a significant drop in classification performance.
> >
> > **(e) Figure 1:** Thanks for pointing this out! We have updated it in the revised paper.
> >
> >
> > **References:**
> >
> > [1] Sudhakaran+, Gate-Shift Networks for Video Action Recognition, CVPR 2020
> >
> > [2] Martinez+, Action recognition with spatial-temporal discriminative filter banks, ICCV 2019
> >
> > [3] Jiang+, STM: SpatioTemporal and Motion Encoding for Action Recognition, ICCV 2019
> >
> > [4] Kwon+, MotionSqueeze: Neural Motion Feature Learning for Video Understanding, ECCV 2020
> >
> > [5] Meng et al., AR-Net: Adaptive Frame Resolution for Efficient Action Recognition, ECCV 2020
> >
> > [6] Bengio, Yoshua, Nicholas Léonard, and Aaron Courville. "Estimating or propagating gradients through stochastic neurons for conditional computation." arXiv preprint arXiv:1308.3432 (2013).

---

> > > ### Comment · AnonReviewer4 · 2020-11-25
> > > **Further discussion points if time allows**
> > >
> > > I appreciate the authors' effort for the additional experiment, the clarification on the questions, and timely update of the paper.
> > >
> > > I understood **(b) Necessity of Gumbel Softmax** and **(c) Output type of the policy net**.
> > > I think the new texts in page 5 are now clear enough to tell how the policy networks behave in forward and backward pass.
> > > Thank you for the clarification.
> > >
> > > Regarding **(d) Reusing multiple times**, I have a further question.
> > > Suppose the situation where the policy is "reuse" on both frame t and t-1, and "keep" on frame t-2. In this case, it is clear that frame t-1 reuses the features of frame t-2, which are already computed at t-2. Does the features on frame t-1 need to be newly computed just for being "reused" in frame t, even though frame t-1 itself does not use the computed features?
> > >
> > > Regarding **(a)  Comparison with recent methods**, which is the most important point, I am still not completely convinced.
> > > I understand the papers I listed are not particularly for efficient action recognition, but it cannot be the reason for not listing them in the comparison tables and figures, especially in Figure 2.
> > >
> > > Probably the authors worry about comparing with inefficient but high-accuracy methods, which may lead to unfair comparison.
> > > However, the comparison to those methods in the plot of Figure 2 stays fair because it has two axis for accuracy and efficiency (FLOPS, by the way I found a typo in the label of the axis "GLOPS").
> > > In this view, the current version of Figure 2 looks rather selective, listing only the methods that make the proposed method look "the best".
> > > I am afraid that the readers may mis-understand that there is no method that has better accuracy at the sacrifice of efficiency.
> > >
> > > Some of the related results are as follows.
> > > In [1], their GSM seems to have better accuracy-efficiency trade off such as (47.24%, 16.46 GFLOPS), (49.01%, 26.85 GFLOPS) compared to AdaFuse (44.9%, 19.1 GFLOPS), (46.8%, 31.3 GFLOPS).
> > > In [2], though they do not have FLOPS count in the paper, they achieved 45% using ResNet18 and 50.1% using ResNet 50.
> > > In [3], they achieved (47.7%, 32.93 GFLOPS).
> > > See also Figure 7 in [1] for your reference.
> > >
> > > Combined with these points, I think the 2nd claim on the contribution in page 2 is too much.
> > > In my view, the proposed method is not entirely model-agnostic, but it is basically effective only for 2D CNN based models.
> > > More specifically, it is effective on top of the models that do not take the temporal information into account by themselves.
> > > If a base model already takes the temporal information into account such as TSM, the improvement brought by the proposed method seems to be marginal.
> > > I admire the advantage of the proposed method that it is expected to benefit from future development of 2D CNN.
> > >
> > > I do not expect further modification in this very short term, but would like to listen to the authors' opinion on those points if possible.

---

> > > > ### Author Response · Authors · 2020-11-25
> > > > **Thanks! We have revised the paper based on your suggestions.**
> > > >
> > > > Dear Reviewer 4,
> > > >
> > > > Thanks again for your constructive review, which has helped us to improve the quality and clarity of the paper. Below are our responses to your concerns and we have updated the paper to reflect these changes.
> > > >
> > > > **(d) Reusing multiple times:** Yes, you are right. The features on frame t-1 need to be newly computed just for being "reused" in frame t.
> > > >
> > > > **(e) Comparison with recent methods:** As we pointed earlier, our main goal is to reduce the computation cost of existing architectures (improve efficiency) while achieving a better classification accuracy and the works in [1-3] are not about efficient action recognition since they usually focus on improving the classification accuracy by adding extra operations. We did not include these methods earlier in Figure 2 as comparison with high-accuracy methods without consideration of efficiency, may lead to unfair comparison. However, as per your suggestion, we now have updated Figure 2 by adding the performance of [3] which uses ResNet50 architecture similar to Ours. We are unable to include the performance of [2] as they don’t report FLOPS in their work. We also believe adding the performance of [1] in Figure 2 will be unfair and misleading as they use different backbone architectures such as Inception V3 to achieve better performance compared to the ResNet50 architecture used in our best method. Besides, [1] is only implemented on Inception-based architectures, where ours can potentially combine with any sort of 2D CNN-based architectures and backbones. Moreover, since our approach is closely related to adaptive video inference models, we have now added the performance of AR-Net in the Figure 2, which shows that AdaFuse significantly outperforms AR-Net in both accuracy and efficiency with 40% less parameters (Table 4 in the revised paper).
> > > >
> > > > Thanks for the clarification on 2D CNN architectures. As you rightly pointed, our approach is model agnostic for 2D CNN architectures with different backbones and we indeed believe this will truly benefit from future development of efficient 2D CNNs for action recognition. We have updated the contributions in the revised paper. Regarding TSM, we observe that the performance of adaptive temporal fusion depends on the position of shift modules in TSM and optimizing the position of such modules through additional regularization could help us not only to achieve better accuracy but also to lower the number of parameters; we leave this as an interesting future work. We have added this discussion in the revised paper. We hope we clarified your concerns and thanks again for your suggestions that helped us to improve the quality of the paper.

---

### Official Review · AnonReviewer3 · 2020-10-31
**A good paper that needs some work**

**Rating:** 7
**Confidence:** 4

**Review:**

#################################

Summary:

The paper presented an adaptive inference model for efficient action recognition in videos. The core of the model is the dynamic gating of feature channels that controls the fusion between two frame features, whereby the gating is conditioned on the input video and helps to reduce the computational cost at runtime. The proposed model was evaluated on several video action datasets and compared against a number of existing deep models. The results demonstrated a good efficiency-accuracy trade-off for the proposed model.

#################################

Pros:
* The paper has a novel idea (adaptive temporal feature fusion) and addresses an important problem in vision (efficient action recognition).
* Solid experiments on multiple datasets. The analysis of the learned policy is quite interesting.
* Well-written paper

#################################

Cons:
* Limited technical novelty

The idea of building adaptive inference models with a policy network for video classification has been previously explored by Wu et al., Meng et al. and others (e.g., skip part of the model, select a subset of frames, choose the input resolution to the model). The main technical component of the model is also very similar to the channel gating network (Hua et al.). The key innovation seems to be the perspective of modeling temporal feature fusion for adaptive inference. This is probably best considered as in parallel to previous approaches for adaptive video recognition. The technical components thus look less exciting.

* Lack of comparison to other adaptive inference models / temporal fusion schemes

There isn’t a real comparison between the proposed method and recent works on adaptive inference video recognition (e.g, Wu et al, Meng et al.). The benefit of model temporal feature fusion --- a main contribution of the paper, thus remains unclear with respect to other design choices (e.g., input resolution or frame selection). I’d suggest some experiments that compare to those work. Another important experiment is to contrast the proposed method with other temporal feature fusion schemes (e.g, LSTM, TSM). For example, TSM --- a hand-crafted feature fusion module, seems to have less number of parameters, slightly higher FLOPs and comparable accuracy (Table 3). If that is the case, the contribution of the proposed adaptive fusion scheme is much weakened.

#################################

Minor comments:

It is not totally clear to me how the FLOPs of the proposed model are computed. As the proposed model will have a different FLOP conditioned on the input video, were the reported FLOPs averaged across the dataset? I was not able to find a description in the paper.

It will be great if the authors can report some run-time performance (e.g., wall time). To achieve the theoretic FLOPs, the proposed model will rely on filter re-arrangement on the fly and sparse convolution kernels. Both can be less efficient on certain devices, e.g., GPUs.


#################################

Justification for score:

All in all a good paper. My main concern is the missing link / comparison to previous works on adaptive video recognition. If this concern can be addressed, I am happy to raise my rating.

---

> ### Author Response · Authors · 2020-11-22
> **Response to AnonReviewer3 (Part 1)**
>
> We thank the reviewer for the insightful questions and great suggestions in terms of adaptive inference models. We have revised the paper by adding the new comparisons with AR-Net (Meng et al. 2020) and LSTM baseline on multiple datasets.
>
> **(a) Technical Novelty:** As rightly pointed by the reviewer, the main idea of our paper is about adaptive temporal fusion which addresses a very important issue in video action recognition. Specifically, the idea of learning a decision policy to dynamically fuse channels from current and history feature maps is novel and largely under addressed in action recognition. Our adaptive fusion model is able to achieve better accuracy with less computation on multiple standard datasets. The prior work ARNet (Meng et al. 2020) [1] simply chooses varied resolution inputs, without really considering the relationship between features from consecutive frames, whereas CGNet (Hua et al. 2019) [2] focuses on spatial pruning which fails to use the temporal correlation to reduce computation for sequential video data. Likewise, AR-Net also fails to capture the important temporal information present in videos, resulting in very poor performance on temporal-rich datasets like Something V1/V2 and Jester (as shown in our new experiments). Our approach, on the other hand, reuses history features when necessary to make the network capable for strong temporal modelling. Moreover, our analysis on learned policy shows that this adaptive temporal fusion can act as an inspector to the dataset and also provide guidance to network architecture designs. Thus we believe our paper has great novelty and impact for both empirical performances, and future works.
>
> **(b) Comparison with adaptive inference models/temporal fusion schemes:** To demonstrate the effectiveness of temporal fusion, we compare our approach with the recent adaptive inference model, namely AR-Net (Meng et al. 2020) [1] on Something-Something-V1, Jester and Mini-Kinetics datasets. Below are the results on different datasets (Table 4 in the revised paper).
>
> $$\\begin{array} {|lrrrr|}
> \\hline \\text{Method (ACC/FLOPS)} & \\text{Params} & \\text{Something V1} & \\text{Jester} & \\text{Mini-Kinetics} \\\\
> \\hline \\text{AR-Net (Meng et al. 2020)} & \\text{63.0M} & \\text{18.9/41.4G} &  \\text{87.8/21.2G} & \\text{71.7/32.0G} \\\\
> \\text{AdaFuse (TSN-ResNet50)} & \\text{37.8M} & \\text{41.9/22.1G} & \\text{94.7/23.0G}  & \\text{72.3/23.0G} \\\\
> \\hline
>  \\end{array}$$
>
> As can be seen, AdaFuse consistently outperforms AR-Net on all three datasets in both
> accuracy and efficiency, while using about 40% less parameters. As expected, AdaFuse obtains the best improvements on temporal rich datasets like Something-Something V1 and Jester, showing its potential for strong temporal modeling in addition to reducing computation for efficient recognition.
>
> Furthemore, as per your suggestion, we contrast our proposed method with a temporal feature fusion baseline, that uses LSTM to update per-frame predictions through hidden states and then averages all predictions as the video-level prediction. Following are the results of LSTM baseline using both ResNet18 and ResNet50 on all four datasets.
>
> $$\\begin{array} {|lrrrr|}
> \\hline \\text{Method (ACC/FLOPS)} & \\text{Something V1} & \\text{Something V2} & \\text{Jester} & \\text{Mini-Kinetics} \\\\
> \\hline \\text{LSTM (TSN-ResNet18)} & \\text{28.4/14.7G} & \\text{40.3/14.7G} &  \\text{93.5/14.7G} & \\text{67.2/14.7G} \\\\
> \\text{AdaFuse (TSN-ResNet18)} & \\text{36.9/10.3G} & \\text{50.5/11.2G} & \\text{93.7/7.6G}  & \\text{67.5/11.8G} \\\\
> \\hline
> \\text{LSTM (TSN-ResNet50)} & \\text{30.1/33.0G} & \\text{47.4/33.0G} & \\text{93.7/33.0G}  & \\text{71.6/33.0G} \\\\
> \\text{AdaFuse (TSN-ResNet50)} & \\text{41.9/22.1G} & \\text{56.8/18.1G} & \\text{94.7/16.1G}  & \\text{72.3/23.0G} \\\\
> \\hline
>  \\end{array}$$

---

> > ### Author Response · Authors · 2020-11-22
> > **Response to AnonReviewer3 (Part 2)**
> >
> > In both cases, our approach, AdaFuse consistently outperforms the LSTM baseline with 35% savings in FLOPS on average. Our approach harvests large gains in accuracy and efficiency on Something V1 & V2, showing the effectiveness of adaptive temporal fusion over LSTM baseline. While LSTM is competitive on both Jester and Mini-Kinetics, AdaFuse still outperforms it while providing 49% and 52% reduction in FLOPs on both datasets respectively. We have added these new results in Table 1 and Table 3 of the revised version. Furthemore, while using adaptive temporal fusion in all the blocks of “TSM” (Lin et al., 2019), AdaFuse achieves more than 40% savings in computing budget but only at 0.7%-0.8% loss in accuracy. As TSM already uses temporal shift operation, we observe too much temporal fusion causes performance degradation due to a worse spatial modelling capability, consistent with the findings in (Lin et al., 2019). On the other hand, using adaptive temporal fusion only in the last block outperforms TSM with a 5% saving in FLOPS (Table 5) on both datasets. From our experiments, we observe that the performance of adaptive temporal fusion depends on the position of shift modules in TSM and optimizing the position of such modules through additional regularization could help us not only to achieve better accuracy but also to lower the number of parameters. We leave this as an interesting future work. We have included a discussion on this in the revised version.
> >
> > **(c) FLOPS computation:** Yes. We are computing the averaged FLOPS across the dataset and have made it clear in our revised version.
> >
> > **(d) Runtime/Hardware:** Thanks for the insightful question on hardware! Sparse convolutional kernels are indeed less efficient on current hardwares, e.g., GPUs. However, we strongly believe it is important to explore models for efficient video recognition which might guide the direction of new hardware development in the years to come. Furthermore, we also expect wall-clock time speed-up in the inference stage via efficient CUDA implementation, which we anticipate will be developed. We have updated the paper to include this discussion in the revised version.
> >
> > **References:**
> >
> > [1] Meng et al., AR-Net: Adaptive Frame Resolution for Efficient Action Recognition, ECCV 2020
> >
> > [2] Hua et al., Channel gating neural networks, NeurIPS 2019

---

> > > ### Comment · AnonReviewer3 · 2020-11-24
> > > **Updated Review**
> > >
> > > I thank the authors for providing additional results and for clarifying the technical details. Most of my previous concerns have been addressed. I believe this is now a solid paper and would recommend its acceptance.

---

### Official Review · AnonReviewer2 · 2020-11-03
**This paper presents an approach to dynamically fuse channels from current and past feature maps for strong temporal modeling of human activities. The authors claim that using a skipping operation reduces the computational complexity of action recognition.**

**Rating:** 7
**Confidence:** 5

**Review:**

Authors assessed how their adaptive temporal fusion network performs on public datasets such as Something V1&2, Kinetics, etc.. The contribution of this paper is in proposing an approach to automatically determine which channels to keep, reuse, or skip per layer and per target instance that can result in efficient action recognition.

STRENGTHS:
The proposed method is model-agnostic, making it easy to use as a plugin operation for other network architectures.
Reusing history features when necessary to make the network capable for strong temporal modeling.

CONCERNS:
The paper has examined the temporal fusion module on BN-Inception and ResNet models, while more recent models’ evaluation is missing.
While the policy network is defined as two FC layers and a ReLU, it is not clear why the authors chose this architecture and how they have tuned it?
In section 3, Using 2D-CNN for Action Recognition, a citation to one of the recent works in modeling the temporal causality is missing: Asghari-Esfeden, Sadjad, Mario Sznaier, and Octavia Camps. "Dynamic Motion Representation for Human Action Recognition." In The IEEE Winter Conference on Applications of Computer Vision, pp. 557-566. 2020.

---

> ### Author Response · Authors · 2020-11-22
> **Response to AnonReviewer2**
>
> We thank the reviewer for the thoughtful reviews and below are our responses to the concerns regarding the more recent backbone and policy network.
>
> **(a) Evaluation on more recent models:** Our adaptive temporal fusion module is model agnostic and hence can be easily plugged into any existing 2D-CNN models. Besides BN-Inception and ResNet models, to investigate the power of AdaFuse on more advanced models, we further implement our idea on top of the EfficientNet [1] architecture (b0, b1, b2, b3) and tested on Something-Something-V1 dataset. The results are as follows.
>
> $$\\begin{array} {|lrrrr|}
> \\hline \\text{Method} & \\text{\\#Params} & \\text{FLOPS} & \\text{Top1} & \\text{Top5} \\\\
> \\hline
> \\text{TSN (Eff-b0)} & \\text{5.3M} & \\text{3.1G} & \\text{18.0} & \\text{44.9} \\\\
> \\text{AdaFuse (TSN-Eff-b0)} & \\text{9.3M} & \\text{2.8G} & \\text{39.0} & \\text{68.1} \\\\
> \\hline
> \\text{TSN (Eff-b1)} & \\text{7.8M} & \\text{5.6G} & \\text{19.3}  & \\text{45.9} \\\\
> \\text{AdaFuse (TSN-Eff-b1)} & \\text{12.4M} & \\text{4.9G} & \\text{40.3}  & \\text{69.2} \\\\
> \\hline
> \\text{TSN (Eff-b2)} & \\text{9.2M} & \\text{8.0G} & \\text{18.8} & \\text{46.0} \\\\
> \\text{AdaFuse (TSN-Eff-b2)} & \\text{13.8M} & \\text{7.2G} & \\text{40.2} & \\text{69.5} \\\\
> \\hline
> \\text{TSN (Eff-b3)} & \\text{12.0M} & \\text{14.4G} & \\text{19.3}  & \\text{46.6} \\\\
> \\text{AdaFuse (TSN-Eff-b3)} & \\text{16.6M} & \\text{12.9G} & \\text{40.7}  & \\text{69.7} \\\\
> \\hline
>  \\end{array}$$
>
>
> Results show that AdaFuse can still bring significant improvement in accuracy and efficiency even on more advanced network architectures, which shows its power in efficient action recognition. We have updated these results in Table 2 of our revised paper.
>
> **(b) Design of the policy network:** The role of the policy network in our approach is to automatically determine which channels of the convolutional feature maps to keep, reuse or skip per layer and per target instance for efficient action recognition. More specifically, given a convolutional layer with N channels in the output feature maps, our policy network predicts a 3*N matrix and uses Gumbel Softmax to further sample the choices (keep, skip or reuse) for those N channels. Fully-connected network can well serve this requirement. Besides, the policy network should also be designed as a lightweight module to avoid massive computation overhead. Thus, we choose a 2-layer fully-connected network as our policy network, using ReLU to bring in the non-linearity and enhance the model capacity. We initialize policy network’s weights randomly, and jointly train them with our backbone networks (TSN/TSM with ResNet/BN-Inception/EfficientNet) guided by both accuracy loss and efficient loss, as shown in Equation (6) of the main paper, which turns out to be working well in our experiments on multiple datasets.
>
> **(c) Missing reference:** Thanks for pointing us to the recent paper on human action recognition. In the revision, we have added the missing reference [2] as suggested.
>
> **References:**
>
> [1] Tan, Mingxing, and Quoc V. Le. "Efficientnet: Rethinking model scaling for convolutional neural networks." ICML. 2019.
>
> [2] Asghari-Esfeden, Sadjad, Mario Sznaier, and Octavia Camps. "Dynamic Motion Representation for Human Action Recognition." In The IEEE Winter Conference on Applications of Computer Vision, pp. 557-566. 2020.

---

### Author Response · Authors · 2020-11-21
**Thanks to all the reviewers. Summary of changes and paper revision.**

We would like to thank all the reviewers for their constructive comments! We are encouraged that reviewers find that our work addresses an important problem of efficient action recognition in vision (R3), our idea on adaptive temporal fusion to be novel and interesting (R1, R3, R4), achieves good accuracy with reasonable computational budget (R4), and is model agnostic, making it easy to use as a plugin operation for other network architectures (R2).

We have addressed all the reviewer’s concerns and incorporated the additional experiments and suggestions into the updated PDF (changes are highlighted in purple). Below,  we summarize the main changes to the paper and encourage the reviewers to take a look at the new additions:
1. Added results using recent architecture (EfficientNet), as suggested by R2 (Table 2).
2. Added comparison with the recent adaptive inference model (AR-Net, Meng et al. 2020), as suggested by R3 (Table 4)
3. Added comparison with LSTM baseline, as suggested by R3 (Table 3)
4. Updated related work with missing references, as suggested by reviewers.

---

### Decision · Program_Chairs · 2021-01-07
**Final Decision**

**Decision:**

Accept (Poster)

**Comment:**

This paper presents a model for video action recognition.  The reviewers appreciated the development of a novel dynamic fusion method that examines channels from feature maps for use in temporal modeling.  After reading the authors' responses, the reviewers converged on an accept rating.  The solid empirical results and analysis, the fact that is is a plug-in method that could be used in other models, and the clear exposition were deemed to be positives.  As such, this paper is accepted to ICLR 2021.